# Growth Patterns of Neonates Treated with Thermal Control in Neutral Environment and Nutrition Regulation to Meet Basal Metabolism

**DOI:** 10.3390/nu11030592

**Published:** 2019-03-11

**Authors:** Shiro Kubota, Masayoshi Zaitsu, Tatsuya Yoshihara

**Affiliations:** 1Kubota Life Science Laboratory Co., Ltd., Saga 840-0535, Japan; kubotahp@gmail.com; 2Kubota Maternity Clinic, Fukuoka 810-0014, Japan; 3Department of Social and Behavioral Sciences, Harvard T.H. Chan School of Public Health, Boston, MA 02115, USA; m-zaitsu@m.u-tokyo.ac.jp; 4Department of Public Health, Graduate School of Medicine, The University of Tokyo, Tokyo 113-0033, Japan; 5Clinical Research Center, Fukuoka Mirai Hospital, Fukuoka 813-0017, Japan

**Keywords:** growth chart, breastfeeding, physiological body-weight loss, thermal control, basal maintenance expenditure

## Abstract

Little is known about the growth patterns of low birth weight neonates (<2500 g) during standardized thermal control and nutrition regulation to meet basal metabolism requirements compared to those of non-low birth weight neonates (2500 g and above). We retrospectively identified 10,544 non-low birth weight and 681 low birth weight neonates placed in thermo-controlled incubators for up to 24 h after birth. All neonates were fed a 5% glucose solution 1 h after birth and breastfed every 3 h (with supplementary formula milk if applicable) to meet basal metabolism requirements. Maximum body-weight loss (%), percentage body-weight loss from birth to peak weight loss (%/day), and percentage body-weight gain from peak weight loss to day 4 (%/day) were assessed by multivariable linear regression. Overall, the growth curves showed a uniform J-shape across all birth weight categories, with a low mean maximum body-weight loss (1.9%) and incidence of neonatal jaundice (0.3%). The body-weight loss patterns did not differ between the two groups. However, low birth weight neonates showed significantly faster growth patterns for percentage body-weight gain: β = 0.52 (95% confidence interval, 0.46 to 0.58). Under thermal control and nutrition regulation, low birth weight neonates might not have disadvantages in clinical outcomes or growth patterns.

## 1. Introduction

During the first days of life, infants show a physiological phenomenon of temporary body-weight loss. However, excess body-weight loss in the neonatal period tends to increase risks of hypoglycemia, hypernatremia, and hyperbilirubinemia, resulting in permanent neurological damage [1]. While the American Academy of Pediatrics criteria currently recommend a cut-off point of 7% for physiological body-weight loss [2], excess loss resulting from inadequate intake is an emerging concern among high-risk populations, such as (but not limited to) East-Asian neonates, even after controlling for genetic polymorphisms [3,4,5,6].

In a cohort of non-low birth weight (NLBW) neonates (birth weight 2500 g and above) in Japan not admitted to the neonatal intensive care unit (NICU), we previously reported a potential standardized neonatal regimen, a local neonatal protocol involving a combination of thermal control and nutrition regulation to meet basal metabolism requirements (~50 kcal/kg), in order to reduce the incidence of neonatal jaundice by preventing excess body-weight loss [7]. This combination method included a neutral thermal environment during the first hours after birth with an ambient room temperature of ~34 °C (93.2 °F) [8,9,10]. In contrast to ambient room temperature (~24 °C/75.2 °F), a neutral thermal environment maintained an optimal body temperature through circulatory stability and improved digestive function [8,9,10]. The combination method also included nutrition regulation to meet the basal metabolism requirement [11], in which neonates were fed a 5% glucose solution 1 h after birth, a convenient method to prevent hypoglycemia and neonatal jaundice [12], followed by breastfeeding every 3 h with supplementary formula milk if applicable. In previous findings, compared to either thermal control or nutrition regulation alone, the combination method had better safety profiles for neonatal growth (e.g., serum glucose levels, circulatory stability, digestive functions, and metabolism) [7,8].

In addition to gestational age and maternal factors (such as maternal smoking, preeclampsia, and gestational diabetes), birth weight plays a crucial role in neonatal body weight growth. Compared to NLBW neonates, low birth weight (LBW) neonates (birth weight <2500 g) are more likely to be at risk for disadvantages in clinical outcomes or growth patterns, such as excess body-weight loss, neonatal jaundice, and delayed body-weight growth [13,14,15,16,17]. However, few studies have focused on LBW neonates not admitted to the NICU, and little is known about the contribution of our combination method among LBW neonates.

Accordingly, the goal of the present study was to characterize the body-weight growth patterns and clinical outcomes in this population of neonates who had received the combination method. Using data from a cohort with NLBW and LBW neonates treated by the combination method, we examined the detailed patterns of body-weight growth among NLBW and LBW neonates during the first days of life. We also examined whether the body-weight growth pattern in LBW neonates differed from that in NLBW neonates under the combination method.

## 2. Materials and Methods

### 2.1. Study Settings

This retrospective, longitudinal study of neonates during the initial days of life used clinical data from a cohort of neonates born in Kubota Maternity Clinic in Fukuoka, Japan (1989 to 2017). The details of the database are described elsewhere [7]. Briefly, Kubota Maternity Clinic was a general obstetrics and gynecological hospital that provided general obstetrics care and closed in 2017. The characteristics of the population were similar to those reported in previous studies in Japan [3,18,19]. All neonates were treated with the combination method (standardized thermal control and nutrition regulation) and discharged by a physician on day 4 or later, when the serum bilirubin level was <15 mg/dL, body weight was recovered from its lowest level, and phototherapy was no longer required. Since the bilirubin level usually peaks at approximately day 4 among Japanese neonates [18,19], discharge on day 4 or later is the standard practice in Japan. Clinical information on all neonates and their mothers had been collected since 1989, except for those who had neonatal asphyxia, congenital heart failure or malformation, or were transferred to a tertiary hospital (such as a university hospital) for neonatal intensive care. We obtained a de-identified dataset through a research agreement with the hospital; the study was approved by the research ethics committees of Hakata Clinic and registered at UMIN-CTR (UMIN000030011).

We included all 11,445 eligible term and preterm neonates treated with the combination method who (1) had a birth weight 1700 g and greater, (2) had an Apgar score of 7 or higher, and (3) were not admitted to the NICU. We excluded 221 neonates with incomplete data (1.9%), yielding a study population of 11,224 neonates.

### 2.2. Thermal Control and Nutrition Regulation to Meet Basal Metabolism

The details of standardized thermal control and nutrition regulation to meet basal metabolism, a local neonatal regimen, have been described elsewhere [7,8]. Briefly, following delivery in a delivery room maintained at ~25 °C (77 °F), neonates were immediately wiped with cotton towels, administered intraoral suction on a warm bed at ~40 °C (104 °F), hugged by their mothers (skin-to-skin contact), and placed in a transparent thermo-controlled incubator (N-ideal H-2000, Nakamura Medical Industry, Tokyo, Japan) within 2 min of birth. Transparent thermo-controlled incubators were placed next to delivery beds in the delivery room (not placed in the NICU). Neonates remained visible and were continuously observed, not only by physicians and nurses (e.g., Apgar score), but also by their mothers. All neonates remained in incubators for at least 2 h. For the first hour, the incubators were set at 34 °C (93.2 °F); for the second hour, the temperature was turned down to 30 °C (86 °F) to help neonates adapt to normal room temperature. After the initial 2 h, NLBW neonates were transferred to a bed in a standard monitoring room set at ~24 to 26 °C (75.2 to 78.8 °F). LBW neonates with birth weights of 2000 to 2499 g stayed in incubators for an extra 2 to 12 h, while those with birth weights <2000 g stayed in incubators for an extra 12 to 24 h at 28 °C (82.4 °F) before transferring to a standard monitoring room. The optimal profiles of the neutral environment for thermogenesis, circulatory stability, and actual central/peripheral body temperatures have been validated in previous studies with human neonates [8,9,10].

For nutritional regulation, neonates were orally fed a 5% glucose solution (10 mL/kg) with 1 mL of vitamin K syrup containing 2 mg menatetrenone 1 h after birth and then breastfed every 3 h [7,12]. If breast milk production was insufficient, neonates were additionally bottle-fed formula milk until they were sated to maintain a 50 kcal/kg basal metabolism [11]. The oral 5% glucose solution has been shown to be a convenient method to prevent hypoglycemia, while allowing continuous breastfeeding [7,12]. In addition, supplemental bottle-feeding with formula milk was based on previous findings suggesting that the estimated calories provided by breast milk were below the basal metabolism a few days after birth among the study population and that nutrition regulation did not affect breastmilk production [7]. For NLBW neonates, high-calorie formula milk (16 kcal/20 mL) was administered for the first 48 h, followed by normal-calorie formula milk (13 kcal/20 mL) after 48 h. For LBW neonates, high-calorie formula milk was administered throughout the hospital stay.

### 2.3. Birth Weight Categories

Neonates, whose birth weights were normally distributed (Figure 1), were grouped into NLBW (2500 g and above; *n* = 10,544) and LBW (<2500 g; *n* = 680) neonates. Both groups were further categorized into a narrower range of birth weights: among LBW neonates, <2000 g (*n* = 20) and 2000 to 2499 g (*n* = 660); among NLBW neonates, 2500 to 2999 g (*n* = 5001), 3000 to 3499 g (*n* = 4752), 3,500 to 3999 g (*n* = 757), and 4000 g and above (*n* = 34).

### 2.4. Assessment of Clinical Outcomes and Changes in Body Weight

Using body-weight measured at birth and every day during the first four days (SW-5200, Toitu, Tokyo, Japan), we determined the percentage body-weight growth every day [((weight each day − birth weight]/birth weight) × 100%) and maximum body-weight loss (([birth weight − minimum weight]/birth weight) × 100%). Excess body-weight loss was defined as a maximum body-weight loss of at least 7% [2]. The incidence of neonatal jaundice was defined as use of phototherapy, which was based on peak bilirubin levels ≥18 mg/dL along with other clinical findings [7].

Due to physiological body-weight loss, we divided the observation period into “decreasing” (before the day of peak weight loss) and “increasing” (after the day of peak weight loss) periods. For each period, we assessed the percentage body-weight change per day. During the first decreasing period from birth to peak weight loss, we determined the body-weight loss per day ((birth weight − minimum weight)/day of peak weight loss) and percentage body-weight loss per day (([birth weight − minimum weight]/birth weight) × (100%/day of peak weight loss)). During the latter increasing period, from peak weight loss to day 4, we determined the body-weight gain per day ((weight at day 4 − minimum weight)/(4 − day of peak weight loss)) and percentage body-weight gain per day ((weight at day 4 − minimum weight]/birth weight) × (100%/(4 − day of peak weight loss)).

### 2.5. Statistical Analysis

To produce growth curves in a growth chart among the 11,674 neonates, we plotted the mean body weights measured at birth and every day during the first four days separately by birth weight categories. We directly connected the mean values with lines, because fitted lines such as splines eliminated the initial drop of body weight in prior analyses (data not shown). We also drew a chart indicating the percentages of body-weight growth stratified by birth weight. In these charts, we additionally plotted the values measured at day 5 for neonates discharged at day 5 and later (10,336 neonates, 92%) and the values measured at one-month neonatal check-ups (approximately 30 days after birth; 8071 neonates, 72%).

The background demographics and clinical outcomes were then compared between the LBW and NLBW neonates by t- or chi-squared tests. In the decreasing period, the regression coefficient (β) and 95% confidence interval (CI) for the percentage body-weight loss per day against birth weight (referent group, NLBW neonates) were estimated by linear regression. Covariates included sex, gestational age, Apgar score, Cesarean delivery, maternal age, maternal body mass index, parity, hypertensive disorders of pregnancy, and birth year. In the increasing period, the same regression analysis was applied for the percentage body-weight gain per day. In stratified analyses using a continuous variable of birth weight, a linear regression model was applied for analysis among NLBW and NLBW neonates, respectively, to check the association between birth weight and percentage body-weight-growth.

The alpha value was set at 0.05 and all P-values were two-sided. Data were analyzed using STATA/MP 13.1 (Stata-Corp, College Station, TX, USA).

## 3. Results

Overall, the growth curves showed a uniform, J-shaped pattern across all birth weight categories (Figure 2). On average, the neonates started to recover their weights within two days of birth and the mean maximum body-weight loss (mean (SD)) was low (1.9% (1.5%)). The incidence of neonatal jaundice was 0.3% and the incidence of excess body-weight loss was 0.4% (Table 1). We did not observe any neonatal morbidity or mortality events, as well as readmission after discharge, during the study period. Most of the background distributions differed between the NLBW and LBW neonates, except for maternal age (Table 1).

Clinical outcomes, including maximum body-weight loss, bilirubin levels, and phototherapy, did not differ between NLBW and LBW neonates. Although the mean body-weight loss per day (grams) differed, the mean body-weight gain per day (grams) did not differ between groups (Table 1). Compared to those in NLBW neonates, the duration to peak weight loss was shorter (1.4 versus 1.3 days, *p* < 0.003) and the percentage of body-weight gain per day was higher (1.4 versus 1.7%, *p* < 0.001) in LBW neonates (Table 1). As birth weight decreased, the percentage body-weight gain increased (Figure 3).

Regression analyses in the decreasing period revealed that, compared to that in NLBW neonates, the percentage body-weight loss per day did not differ in LBW neonates (Table 2). In the increasing period, compared to that in NLBW neonates, the percentage body-weight gain per day was significantly higher in LBW neonates (Table 2): β = 0.52 (95% CI, 0.46 to 0.58). In stratified analyses in the decreasing period, a higher percentage of body-weight loss per day was only associated with higher birth weights in NLBW neonates (Table 3). In the increasing period, a higher percentage of body-weight gain per day was significantly associated with lower birth weights across all birth weight categories (Table 3).

## 4. Discussion

Under a local neonatal regimen involving a standardized thermal control and nutrition regulation to meet basal metabolism, neonatal growth curves showed a uniform, J-shaped pattern with an initial physiological body-weight loss (~2% loss from birth weight), regardless of birth weight. The incidence of excess body-weight loss was low (<1%), with a short duration to peak weight loss (<2 days) and a low incidence of neonatal jaundice (0.3%). Compared to NLBW neonates, LBW neonates did not appear to have disadvantages in clinical outcomes, including excess body-weight loss and neonatal jaundice. LBW neonates also potentially showed faster body-weight gain patterns, with a trend suggesting that a higher percentage body-weight gain per day was associated with lower birth weight.

In the contemporary understanding of neonatal body-weight growth, the gradient of birth weight for the speed of body-weight gain is positive. That is, heavier birth weight neonates enjoy the advantages of faster body-weight gain and neonatal body-weight growth curves have continuously shown a divergent pattern across birth weight categories [13,14,15,16,17]. During the second half of the twentieth century, when perinatal and neonatal medicine and NICU were on the rise, this positive gradient in birth weight for neonatal body-weight gain was considered a rule of thumb and early illustrations of this positive gradient date as far back as Dancis et al. in 1948 [13]. Over the course of the twentieth century, this positive gradient persisted, reflecting larger insensible water loss in lower birth weight neonates (more premature neonates) [15,17,20]. Part of the reason for our observed pattern, which may potentially buck the contemporary trend, may be due to the neutral thermal environment that improves digestive function with circulatory stability, resulting in the neonatal capability to be fed sufficiently to meet basal metabolism and prolonged thermal regulation in LBW neonates [8,9,10,11,12]. Additionally, in a recent study with exclusively breastfed neonates, 50% of neonates recovered their weight around 10 days after birth [21]. By contrast, most of the neonates recovered their weight by 4 days after birth in our study, suggesting that our nutrition regulation with supplemental formula milk may successfully meet the basal metabolism in our high-risk population.

Body weight is a robust and straightforward anthropometric indicator to grow neonates. The criteria for body-weight-growth have been varied, which may partly due to differences in growth patterns of breastfeeding and formula-feeding based on calorific intakes. For example, in Western countries, studies suggest the potential cut-off point of ~5% to 12% for exclusively breastfed neonates and ~2% to 3% for formula-fed neonates [22,23,24,25,26], while the American Academy of Pediatrics recommends explicitly the cut-off point of 7% for excess body-weight loss among exclusively breastfed neonates [2]. By contrast, in Japan, although some studies suggest the potential cut-off point of ~4% to 10% [3,7], and other expert opinions even support the cut-off point of up to ~15%, any evidence-based clinical guidelines have yet to be established. Given the emerging concern for excess body-weight loss from inadequate intake in East-Asian countries including Japan [3,4,5], physiological body-weight loss would be a body-weight loss for neonates adequately fed to meet basal metabolism, at least in high-risk populations. Indeed, we observed an initial ~2% body-weight loss as the physiological loss with a low incidence of neonatal jaundice in our study population. Therefore, our growth chart may help not only healthcare providers but also mothers of neonates in that high-risk population.

Gestational age and maternal factors during pregnancy may affect neonatal growth pattern. In previous studies, gestational age ≤38 weeks was associated with an increased risk of neonatal jaundice [27]. Maternal smoking during pregnancy and preeclampsia were associated with decreased birth weight, whereas gestational diabetes was associated with increased birth weight [28,29,30]. In Finland, comprehensive screening of gestational diabetes was associated with decreased mean birth weight and macrosomia rates but was associated with an increased prevalence of neonatal hypoglycemia [30]. In our study, gestational age was consistently associated with neonatal growth pattern. However, significant effects of hypertensive disorders of pregnancy and maternal body mass index were inconsistent under the combination method (although we could not assess the impact of gestational diabetes due to the limitation of our data). Additionally, no hypoglycemia was observed in our high-risk population. Therefore, in addition to LBW neonates, neonates with gestational age ≤38 weeks, maternal smoking, preeclampsia, and gestational diabetes, might benefit the most from the combination method.

Some limitations should be noted. First, we did not have a control group with other nutrition treatments (exclusively breastfeeding and exclusively formula-feeding). In addition, we did not have hourly data of body weight and did not include sick infants who required intravenous infusion [13,25,31]. The thermal regulation also differed between birth weight categories, thereby introducing potential bias and limiting external generalizability. However, internal validity might be maintained for the comparison of NLBW and LBW neonates from the same source population. Second, other anthropometric parameters, such as body length and head circumference, and potential maternal confounding variables. such as diabetes, were not available [15,16,17,32]. However, body weight is a reliable indicator of neonatal growth [7,15]. Third, use of supplemental feeding is controversial due to the risk of breastfeeding failure, and excessive weight gain and the combination method might raise concerns for introducing unintended adverse events [1,33]. However, in previous studies, supplemental feeding did not affect breast milk production [7] and breastfeeding was not consistently associated with the reduction of obesity or body fat mass in children or adolescents [34]. In addition, a recent randomized clinical trial concluded that supplementing breastfeeding with early limited formula milk (10 mL) did not interfere with breastfeeding [35]. Fourth, we were not able to assess the impact of standardized early skin-to-skin contact on the incidence of hypothermia or hypoglycemia compared to our early skin-to-skin contact within 2 min. However, in previous studies, profiles of serum glucose levels and core/peripheral distribution of body temperature were likely better in neonates with thermal regulation compared with those without thermal regulation [7,8,9]. Therefore, future studies, including randomized controlled trials, are warranted to further understand how thermal control and nutrition regulation influence neonatal growth.

Lastly, in the era of early neonatal hospital discharge, standardized thermal regulation and nutrition regulation to meet basal metabolism (without NICU care) may reduce medical expenditures of readmission for phototherapy with costly NICU care among LBW neonates [1,36]. Even in our high-risk study population, we observed a very low incidence of neonatal jaundice (0.4%), which corresponds to the incidence in Western settings [27,37]. Given the very low incidence of neonatal jaundice, a potential reduction of medical expenditure would be expected by the combination method, in addition to improved education (clinicians, nurses, and parents) and systemic institutional and community-wide approaches [38,39]. Due to extra thermal regulation that might further stabilize cardiopulmonary circulation and improve digestive function, no phototherapy was required in NLBW neonates [8,9,10]. In addition, the combination method did not require intravenous infusion therapy and did not result in neonatal morbidity, mortality, or readmission after discharge, thereby potentially increasing the efficient use of NICU resources and potentially reducing clinical complications associated with intravenous infusion among LBW neonates. Furthermore, the initial 2 h spent in the incubator might help to avoid the substantially longer incubation required for phototherapy, thereby potentially increasing opportunities for physical contact between mothers and their babies.

In conclusion, neonatal body-weight growth patterns were characterized under standardized thermal control and nutrition regulation, with a low incidence of unfavorable clinical outcomes, regardless of birth weight category. Under this combination method, LBW neonates might not have disadvantages in clinical outcomes or growth patterns.

## Figures and Tables

**Figure 1 nutrients-11-00592-f001:**
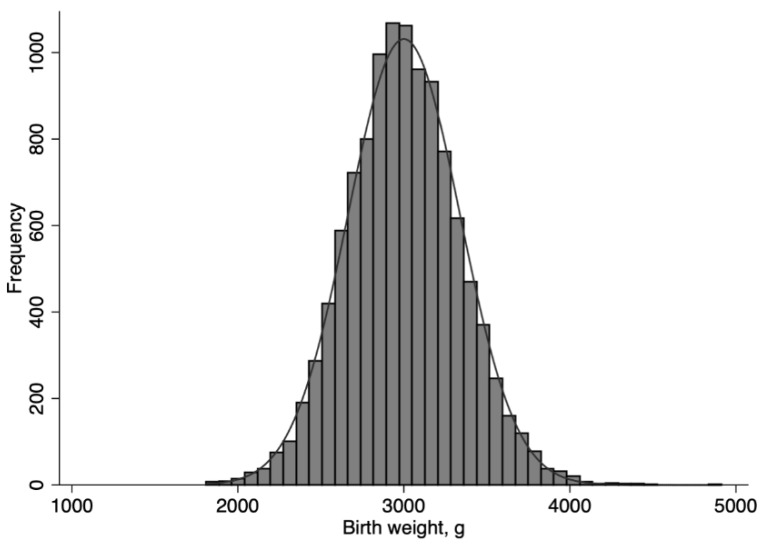
Distribution of birth weight among 11,224 neonates who received standardized thermal control and nutrition regulation to meet basal metabolism.

**Figure 2 nutrients-11-00592-f002:**
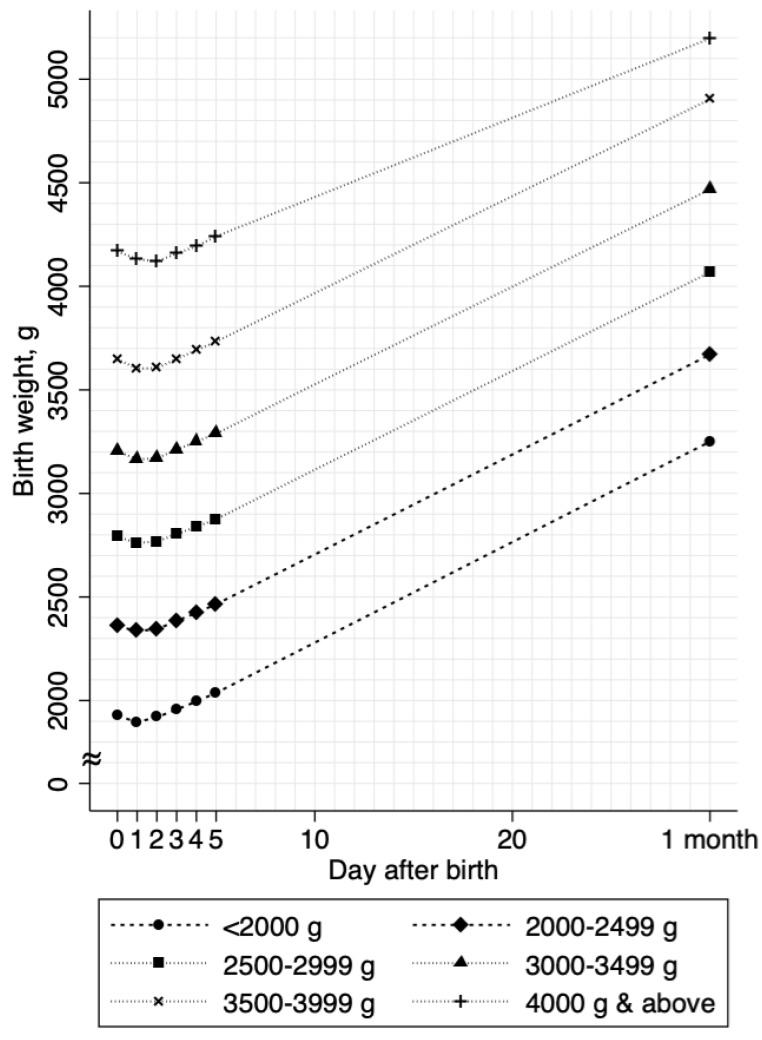
Body weight growth curves stratified by birth weights. The mean body weights are shown as connected lines stratified by birth weight categories. Values from day 0 through day 4 were estimated in 11,224 neonates; values at day 5 and 1 month (approximately 30 days of birth) were estimated in 10,336 neonates and 8071 neonates, respectively.

**Figure 3 nutrients-11-00592-f003:**
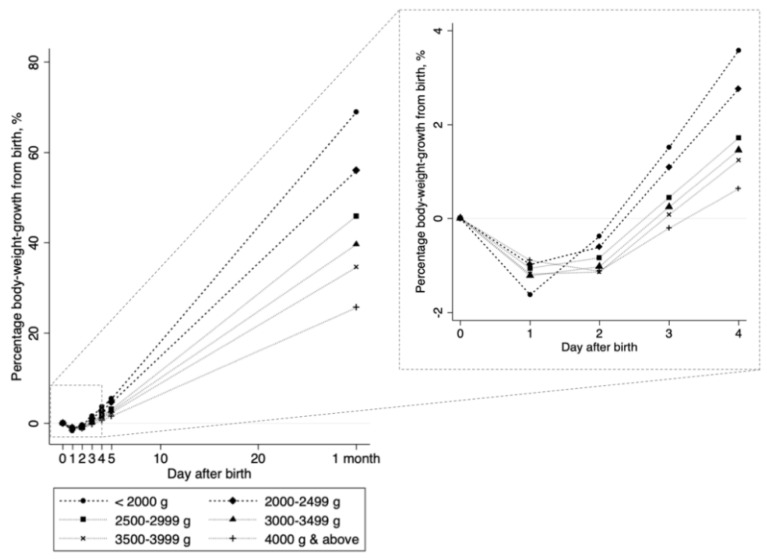
Daily percentages of bodyweight growth against birth weight. Each connected line shows the change in the percentage bodyweight growth from birth weight (([weight at every day − birth weight]/birth weight) × 100%), stratified by birth weight categories. Values from day 0 through day 4 were estimated in 11,224 neonates; values at day 5 and 1 month (approximately 30 days of birth) were estimated in 10,336 neonates and 8071 neonates, respectively.

**Table 1 nutrients-11-00592-t001:** Background and clinical characteristics of 11,224 neonates who received optimal thermal control with sufficient nutrition.

Characteristics	Mean (SD) or number (%)			*p*
	Total *n* = 11,224	NLBW Neonates (≥2500 g) *n* = 10,544	LBW Neonates (<2500 g) *n* = 680	
**Background characteristics**				
Female	5468 (49%)	5066 (48%)	402 (59%)	<0.001
Gestational age (weeks)	39.0 (1.2)	39.0 (1.1)	37.9 (1.4)	<0.001
Birth weight (g)	3000 (337)	3,043 (300)	2346 (136)	<0.001
Apgar score at 1 min	9.6 (0.6)	9.6 (0.6)	9.5 (0.6)	<0.001
Caesarean delivery	899 (8.0%)	810 (7.7%)	89 (13%)	<0.001
Maternal age (years)	31 (4)	31 (4)	31 (4)	0.58
Maternal BMI (kg/m^2^)	20.0 (2.0)	20.1 (2.0)	19.7 (2.0)	<0.001
Multipara	5,256 (47%)	4,982 (47%)	274 (40%)	<0.001
Hypertensive disorders of pregnancy	43 (0.4%)	36 (0.3%)	7 (1.0%)	0.005
Birth year	2001 (8)	2001 (8)	2002 (8)	0.001
**Clinical outcomes**				
Maximum weight loss (%)	1.9 (1.5)	1.9 (1.5)	1.8 (1.5)	0.056
Excess weight loss 7% and above	44 (0.4%)	39 (0.4%)	5 (0.7%)	0.14
Day of peak weight loss	1.4 (0.9)	1.4 (0.9)	1.3 (0.9)	0.004
Body-weight loss per day (g)	40 (36)	41 (36)	31 (29)	<0.001
Percentage body-weight loss per day (%) ^1^	1.3 (1.2)	1.3 (1.2)	1.3 (1.2)	0.52
Body-weight gain per day (g)	41 (24)	41 (25)	39 (21)	0.06
Percentage body-weight gain per day (%) ^2^	1.4 (0.8)	1.4 (0.8)	1.7 (0.9)	<0.001
Peak bilirubin level at day 4 (mg/dL)	8.5 (2.7)	8.5 (2.7)	8.4 (2.7)	0.42
Phototherapy	30 (0.3%)	30 (0.3%)	0 (0.0%)	0.16

BMI, body mass index; NLBW, non-low birth weight; LBW, low birth weight. ^1^ Defined as ((birth weight − minimum weight)/birth weight) × (100%/day of peak weight loss). ^2^ Defined as ((weight at day 4 − minimum weight)/birth weight) × (100%/(4 − day of peak weight loss])).

**Table 2 nutrients-11-00592-t002:** Body weight growth patterns against birth weight estimated by multivariable linear regression analysis.

Characteristics	Model 1 ^3^		Model 2 ^4^	
	β (95% CI)	*p*	β (95% CI)	*p*
**Percentage body weight loss per day** ^1^				
LBW (<2500 g)	−0.03 (−0.12 to 0.06)	0.52	0.03 (−0.06 to 0.13)	0.48
Female			0.05 (0.004 to 0.09)	0.03
Gestational week			0.03 (0.01 to 0.05)	0.001
Apgar score			0.13 (0.09 to 0.17)	<0.001
Caesarean delivery			0.12 (0.03 to 0.20)	0.005
Maternal age			0.003 (−0.003 to 0.01)	0.34
Maternal body mass index			−0.03 (−0.04 to −0.01)	<0.001
Multipara			0.21 (0.16 to 0.26)	<0.001
Hypertensive disorders of pregnancy			0.24 (−0.11 to 0.59)	0.18
Birth year			−0.02 (−0.02 to −0.02)	<0.001
**Percentage body weight gain per day** ^2^				
LBW (<2500 g)	0.33 (0.26 to 0.39)	<0.001	0.52 (0.46 to 0.58)	<0.001
Female			−0.12 (−0.15 to −0.10)	<0.001
Gestational week			0.15 (0.14 to 0.16)	<0.001
Apgar score			−0.02 (−0.04 to 0.01)	0.17
Caesarean delivery			0.001 (−0.05 to 0.05)	>0.99
Maternal age			−0.005 (−0.005 to 0.004)	0.82
Maternal body mass index			0.001 (−0.01 to 0.01)	0.97
Multipara			−0.08 (−0.11 to −0.05)	<0.001
Hypertensive disorders of pregnancy			0.05 (−0.18 to 0.29)	0.66
Birth year			−0.02 (−0.02 to −0.01)	<0.001

β, regression coefficient; CI, confidence interval; LBW, low birth weight. ^1^ Defined as ((birth weight − minimum weight)/birth weight) × (100%/day of peak weight loss). ^2^ Defined as ((weight at day 4 − minimum weight)/birth weight) × (100%/(4 − day of peak weight loss)). ^3^ Simple linear regression model for body-weight growth patterns against birth weight (referent group, NLBW neonates). Data were estimated in 10,544 non-low birth weight neonates and 680 low birth weight neonates. ^4^ Additionally adjusted for potential confounding variables of sex, gestational age, Apgar score, Cesarean delivery, maternal age, maternal body mass index, parity, hypertensive disorders of pregnancy, and birth year.

**Table 3 nutrients-11-00592-t003:** Regression coefficients and 95% confidence intervals estimated with a continuous variable of birth weight in regression analyses, stratified by non-low birth weight and low birth weight neonates.

Characteristics	Percentage Body-Weight-Loss per Day ^1^		Percentage Body-Weight-Gain per Day ^2^	
	β (95% CI) ^3^	*p*	β (95% CI)^3^	*p*
**Overall (*n* = 11,224)**				
Birth weight (per 100 g)	0.01 (−0.003 to 0.01)	0.06	−0.05 (−0.06 to −0.05)	<0.001
**Non-LBW (2500 g and above, *n* = 10,544)**				
Birth weight (per 100 g)	0.01 (0.001 to 0.02)	0.02	−0.04 (−0.05 to −0.03)	<0.001
**LBW (<2500 g, *n* = 680)**				
Birth weight (per 100 g)	0.003 (−0.07 to 0.08)	0.93	−0.16 (−0.21 to −0.11)	<0.001

Abbreviations: β, regression coefficient; CI, confidence interval; LBW, low birth weight. ^1^ Defined as ((birth weight − minimum weight)/birth weight) × (100%/day of peak weight loss). ^2^ Defined as ((weight at day 4 − minimum weight)/birth weight) × (100%/(4 − day of peak weight loss)). ^3^ Multivariable linear regression analysis for body-weight-growth patterns against birth weight, adjusted for sex, gestational age, Apgar score, cesarean delivery, maternal age, maternal body mass index, parity, hypertensive disorders of pregnancy, and birth year.

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
