# Peer review of "Growth Patterns of Neonates Treated with Thermal Control in Neutral Environment and Nutrition Regulation to Meet Basal Metabolism"

_nutrients, 2019, doi:10.3390/nu11030592_

Reviewer 1 Report

This is an interesting study using data from a cohort of neonates who have received standardized thermal control and nutrition regulation to meet basal metabolism. The authors examine the growth-patterns among NLBW and LBW participants during the first days of life. The study outline is novel, however the importance to the field is limited due to the limitations the authors have pointed out.

ABSTRACT

The sentence “All neonates were fed a 5% glucose 21 solution 1 h after birth and breastfed every 3 h with supplementary formula milk” should be clarified.

INTRODUCTION’

Intro is well organized and clear. However I would suggest mentioning briefly also other factors strongly influencing the early growth, such as gestational age, maternal factors (gestational diabetes, pre-eclampsia, maternal smoking during pregnancy).

METHODOLOGY AND RESULTS

Clearly written. Statistical methods used seem correct and logical. The figures provided are clear and informative.

DISCUSSION

The discussion is well-written. The authors however do not go into the effects of gestational age or maternal pregnancy although these may also influence the early growth in addition to early nutrition. I would suggest adding a paragraph discussing all factors related to the early neonatal growth and then discussing the findings of this study in relation to these factors.

Author Response

Response to Reviewer 1 Comments

Reviewer #1

This is an interesting study using data from a cohort of neonates who have received standardized thermal control and nutrition regulation to meet basal metabolism. The authors examine the growth-patterns among NLBW and LBW participants during the first days of life. The study outline is novel, however the importance to the field is limited due to the limitations the authors have pointed out.

We thank the reviewer for this comment. Our point-by-point responses to each of the reviewer’s comments are as follows.

Point 1: ABSTRACT: The sentence “All neonates were fed a 5% glucose 21 solution 1 h after birth and breastfed every 3 h with supplementary formula milk” should be clarified.

Response 1: We thank the reviewer for this comment. We clarified this part as follows:

Abstract, L21:

(Original): “All neonates were fed a 5% glucose solution 1 h after birth and breastfed every 3 h with supplementary formula milk, if applicable”

(Revised, revised parts are underlined): “All neonates were fed a 5% glucose solution 1 h after birth and breastfed every 3 h (with supplementary formula milk if applicable)

Point 2: INTRODUCTION: Intro is well organized and clear. However I would suggest mentioning briefly also other factors strongly influencing the early growth, such as gestational age, maternal factors (gestational diabetes, pre-eclampsia, maternal smoking during pregnancy).

Response 2: We thank the reviewer for this comment. We added brief introduction of other factors influencing neonatal growth as follows:

Introduction, L61:

(Original): “In the contemporary understanding of neonatal body weight growth, compared to NLBW neonates”

(Revised, revised parts are underlined): “In addition to gestational age and maternal factors (such as maternal smoking, preeclampsia, and gestational diabetes), birth weight plays a crucial role in neonatal body weight growth. Compared to NLBW neonates”

Point 3: METHODOLOGY AND RESULTS: Clearly written. Statistical methods used seem correct and logical. The figures provided are clear and informative.

Response 3: We thank the reviewer for this comment.

Point 4: DISCUSSION: The discussion is well-written. The authors however do not go into the effects of gestational age or maternal pregnancy although these may also influence the early growth in addition to early nutrition. I would suggest adding a paragraph discussing all factors related to the early neonatal growth and then discussing the findings of this study in relation to these factors.

Response 4: We thank the reviewer for this comment. We agree that gestational age and maternal factors during pregnancy may affect neonatal growth pattern. In previous studies, gestational age ≤38 weeks was associated with an increased risk of neonatal jaundice [Maisels 1998]. Maternal smoking during pregnancy and preeclampsia were associated with decreased birth weight, whereas gestational diabetes was associated with increased birth weight [Inoue 2017; Jung 2018; Koivunen 2017]. In Finland, comprehensive screening of gestational diabetes was associated with decreased mean birth weight/macrosomia rates but was associated with an increased prevalence of neonatal hypoglycemia [Koivunen 2017]. In our study, we mutually adjusted for low birth weight, gestational age, maternal body mass index, and hypertensive disorders of pregnancy in regression models, and found that gestational age was consistently associated with neonatal growth pattern. However, significant effects of hypertensive disorders of pregnancy and maternal body mass index were inconsistent under the combination method (although we could not assess the impact of gestational diabetes due to the limitation of our data). Additionally, no hypoglycemia was observed in our high-risk population. Therefore, in addition to low-birth weight neonates, neonates with gestational age ≤38 weeks, maternal smoking, preeclampsia, and gestational diabetes, might benefit the most from the combination method.

We added the following paragraph in the discussion section as follows:

Discussion, L298:

(Revised, revised parts are underlined): “Gestational age and maternal factors during pregnancy may affect neonatal growth pattern. In previous studies, gestational age ≤38 weeks was associated with an increased risk of neonatal jaundice [Maisels 1998]. Maternal smoking during pregnancy and preeclampsia were associated with decreased birth weight, whereas gestational diabetes was associated with increased birth weight [Inoue 2017; Jung 2018; Koivunen 2017]. In Finland, comprehensive screening of gestational diabetes was associated with decreased mean birth weight/macrosomia rates but was associated with an increased prevalence of neonatal hypoglycemia [Koivunen 2017]. In our study, gestational age was consistently associated with neonatal growth pattern. However, significant effects of hypertensive disorders of pregnancy and maternal body mass index were inconsistent under the combination method (although we could not assess the impact of gestational diabetes due to the limitation of our data). Additionally, no hypoglycemia was observed in our high-risk population. Therefore, in addition to LBW neonates, neonates with gestational age ≤38 weeks, maternal smoking, preeclampsia, and gestational diabetes, might benefit the most from the combination method.

References

Maisels, M.J.; Kring, E. Length of Stay, Jaundice, and Hospital Readmission. Pediatrics 1998, 101, 995-998.Inoue, S.; Naruse, H.; Yorifuji, T.; Kato, T.; Murakoshi, T.; Doi, H.; Subramanian, S.V. Impact of Maternal and Paternal Smoking on Birth Outcomes. J. Public. Health. (Oxf) 2017, 39, 1-10.

Inoue, S.; Naruse, H.; Yorifuji, T.; Kato, T.; Murakoshi, T.; Doi, H.; Subramanian, S.V. Impact of Maternal and Paternal Smoking on Birth Outcomes. J. Public. Health. (Oxf) 2017, 39, 1-10.

Hung, T.H.; Hsieh, T.T.; Chen, S.F. Risk of Abnormal Fetal Growth in Women with Early- and Late-Onset Preeclampsia. Pregnancy Hypertens. 2018, 12, 201-206.

Koivunen, S.; Torkki, A.; Bloigu, A.; Gissler, M.; Pouta, A.; Kajantie, E.; Vaarasmaki, M. Towards National Comprehensive Gestational Diabetes Screening - Consequences for Neonatal Outcome and Care. Acta Obstet. Gynecol. Scand. 2017, 96, 106-113.

Reviewer 2 Report

Drs Kubota, Zaitsu and Yoshihara describe a model of early neonatal care combining nutritional and thermoregulation support.

The study is based on a significant number of low and non-low birthweight infants, defined as above or below 2500g. 

The outcomes are classified into development of weight over the first 4 days and weight at time of healthy infant examination at one month.

The study has a large number at birth with about 80% of infants being seen at 30 days and detailed data on weight. It spans a long time period, thereby raising the possibility that other aspects of care might have changed over time. The statistical analyses are adequately described. 

Style/language: 

Some parts of the manuscript should be shortened (paragraph L58-66; Discussion-paragraph L243-265) and some stylistic improvements can be made 

L 290: suggest ... might raise concerns...

L 282 (Discussion): clarification: hourly data as to infusion rates or hour specific bilirubin; or hourly temperature measurements? 

Discussion

The authors adequately address the limitations and strengths of the study.

The rate of hyberbilirubinaemia seems to be very low but compatible with earlier reports 

(Eggert LD, et al  Pediatrics 2006;117(5):e855-e862.; Bhutani VK et al   J Obstet Gynecol Neonatal Nurs 2006;35:444-55.; Maisels MJ, Kring EA.  Pediatrics 1998;101:995-8.). All infants presumably  requiring phototherapy were in the normal weight group, the cause of which could be explored. Although it is assumed all jaundiced infants presented within the first 4 days the rate of readmission (if any) if retrievable may also add support to implementation of this protocol.  It seems this has been in part discussed by the authors in their previous publication (Neonatology 2018). 

As this is a single-centre study it is not clear what the incidence of hypothermia /hypoglycemia would be if there were a policy of early skin-to-skin contact. It should also be discussed whether a subgroup would benefit the most from the combination approach (e.g. less than 2500 g and or < than a certain gestational age).

The time to regain birth weight is short, this points to the fact that most babies were probably predominantly formula-fed. Weight changes with exclusive breastfeeding have been described recently (in the Kaiser Permanente Healthcare Network;  Weight Change Nomograms for the First Month After Birth. Paul IM, et al  Pediatrics. 2016 Dec;138(6).

As the authors point out it is not clear if this policy of early introduction of (partial?) formula feeding resulted in reduced numbers of breastfeeding at one month or later. This current study could be added (Flaherman V et al. J Pediatr. 2018 May;196:84-90.) for discussion, which also discusses readmission rates. 

The cost-effectiveness of this model of care should also be put into context. In some jurisdictions a community midwifery programme may supplement a model of early discharge (Flaherman V et al  Health Care Utilization in the First Month After Birth and Its Relationship to Newborn Weight Loss and Method of Feeding. Acad Pediatr. 2018 Aug;18(6):677-684). 

Author Response

Response to Reviewer 2 Comments

Reviewer #2

Drs Kubota, Zaitsu and Yoshihara describe a model of early neonatal care combining nutritional and thermoregulation support. The study is based on a significant number of low and non-low birthweight infants, defined as above or below 2500g. The outcomes are classified into development of weight over the first 4 days and weight at time of healthy infant examination at one month. The study has a large number at birth with about 80% of infants being seen at 30 days and detailed data on weight. It spans a long time period, thereby raising the possibility that other aspects of care might have changed over time. The statistical analyses are adequately described.

We thank the reviewer for this comment. Our point-by-point responses to each of the reviewer’s comments are as follows.

Point 1: Style/language: Some parts of the manuscript should be shortened (paragraph L58-66; Discussion-paragraph L243-265) and some stylistic improvements can be made (L 290: suggest ... might raise concerns...; L 282 (Discussion): clarification: hourly data as to infusion rates or hour specific bilirubin; or hourly temperature measurements?)

Response 1: We thank the reviewer for this comment. We shortened the two paragraphs. Also, we meant hourly data as to hourly body weight data. We revised the text accordingly as follows:

Introduction, L61:

(Original) “In the contemporary understanding of neonatal body weight growth, compared to NLBW neonates, low birth weight (LBW) neonates (birth weight<2,500 g) are more likely to be at risk for disadvantages in clinical outcomes and/or growth patterns, such as excess body-weight loss, neonatal jaundice, and delayed body-weight growth. Hence, potential neonatal regimens in this population should be a priority to tackle the inequalities of LBW neonates in neonatal growth. However, few studies have focused on LBW neonates not admitted to the NICU. In addition, little is known about the contribution of our combination method to body-weight growth patterns and consequent clinical outcomes among LBW neonates not admitted to the NICU.”

(Revised, revised parts are underlined): “In addition to gestational age and maternal factors (such as maternal smoking, preeclampsia, and gestational diabetes), birth weight plays a crucial role in neonatal body weight growth. Compared to NLBW neonates, low birth weight (LBW) neonates (birth weight<2,500 g) are more likely to be at risk for disadvantages in clinical outcomes and/or growth patterns, such as excess body-weight loss, neonatal jaundice, and delayed body-weight growth. However, few studies have focused on LBW neonates not admitted to the NICU, and little is known about the contribution of our combination method among LBW neonates.”

Discussion, L321:

(Original): “the combination method might have concerns for introducing unintended adverse events”

(Revised, revised parts are underlined): “the combination method might raise concerns for introducing unintended adverse events”

Discussion, L313:

(Original): “hourly data”

(Revised, revised parts are underlined): “hourly data of body weight

Point 2: Discussion: The authors adequately address the limitations and strengths of the study. The rate of hyberbilirubinaemia seems to be very low but compatible with earlier reports (Eggert LD, et al. Pediatrics 2006;117(5):e855-e862.; Bhutani VK et al. J Obstet Gynecol Neonatal Nurs 2006;35:444-55.; Maisels MJ, Kring EA. Pediatrics 1998;101:995-8.). All infants presumably requiring phototherapy were in the normal weight group, the cause of which could be explored. Although it is assumed all jaundiced infants presented within the first 4 days the rate of readmission (if any) if retrievable may also add support to implementation of this protocol. It seems this has been in part discussed by the authors in their previous publication (Neonatology 2018).

Response 2: We thank the reviewer for this comment. Even in our high-risk population, we observed a very low incidence of neonatal jaundice (0.4%), which corresponds to the incidence in Western settings [Maisels 1998; Eggert 2006]. We did not observe any infants who required readmission after discharge. Additionally, no infants required phototherapy in the low birth weight group, who were treated with extra thermal regulation. This observed pattern may be mainly attributable to an extra thermal regulation that may further stabilize cardiopulmonary circulation and improve digestive function [Kubota 1988; Yoshimura 2007; Hey 1970]. We revised the text accordingly as follows:

Results, L190:

(Original): “We did not observe any neonatal morbidity or mortality events during the study period.”

(Revised, revised parts are underlined): “We did not observe any neonatal morbidity or mortality events, as well as readmission after discharge, during the study period.”

Discussion, L337:

(Revised, revised parts are underlined): “Even in our high-risk population, we observed a very low incidence of neonatal jaundice (0.4%), which corresponds to the incidence in Western settings [Maisels 1998; Eggert 2006]”

Discussion, L336:

(Original): “The combination method did not require intravenous infusion therapy and did not result in neonatal morbidity or mortality,”

(Revised, revised parts are underlined): “Due to an extra thermal regulation that might further stabilize cardiopulmonary circulation and improve digestive function, no phototherapy was required in NLBW neonates [Kubota 1988; Yoshimura 2007; Hey 1970]. In addition, the combination method did not require intravenous infusion therapy and did not result in neonatal morbidity, mortality, or readmission after discharge,

References:

Maisels, M.J.; Kring, E. Length of Stay, Jaundice, and Hospital Readmission. Pediatrics 1998, 101, 995-998.

Eggert, L.D.; Wiedmeier, S.E.; Wilson, J.; Christensen, R.D. The Effect of Instituting a Prehospital-Discharge Newborn Bilirubin Screening Program in an 18-Hospital Health System. Pediatrics 2006, 117, e855-62.

Kubota, S.; Koyanagi. T.; Hori, E.; Hara, K.; Shimokawa, H.; Nakano, H. Homeothermal adjustment in the immediate postdelivered infant monitored by continuous and simultaneous measurement of core and peripheral body temperatures. Biol Neonate 1988, 54, 79-85.

Yoshimura, T.; Tsukimori, K.; Wake, N.; Nakano, H. The influence of thermal environment on pulmonary hemodynamic acclimation to extrauterine life in normal full-term neonates. J Perinat Med 2007, 35, 236-240.

Hey, E.N.; Katz, G. The optimum thermal environment for naked babies. Arch Dis Child 1970, 45, 328-334.

Point 3: As this is a single-centre study it is not clear what the incidence of hypothermia /hypoglycemia would be if there were a policy of early skin-to-skin contact. It should also be discussed whether a subgroup would benefit the most from the combination approach (e.g. less than 2500 g and or < than a certain gestational age).

Response 3: We thank the reviewer for the comment. In our study, we were not able to assess the impact of a standardized early skin-to-skin contact on the incidence of hypothermia/hypoglycemia compared to our early skin-to-skin contact within 2 min. This is one of the limitations. However, profiles of serum glucose levels and core/peripheral distribution of body temperature were likely better in neonates with thermal regulation compared with those without thermal regulation [Zaitsu 2018; Kubota 1998; Yoshimura 2007].

Neonates with birth weight<2500 g and gestational age ≤38 weeks tend to have risks for consequent adverse clinical outcomes [Maisels 1998], which corresponds to our results. This subgroup would benefit the most from the combination method.

We added these discussions in the text, accordingly as follows:

Discussion, L326:

(Revised, revised parts are underlined): “Fourth, we were not able to assess the impact of standardized early skin-to-skin contact on the incidence of hypothermia/hypoglycemia compared to our early skin-to-skin contact within 2 min. However, in previous studies, profiles of serum glucose levels and core/peripheral distribution of body temperature were likely better in neonates with thermal regulation compared with those without thermal regulation [Zaitsu 2018; Kubota 1998; Yoshimura 2007].

Discussion, L308:

(Revised, revised parts are underlined): “Therefore, in addition to LBW neonates, neonates with gestational age ≤38 weeks, maternal smoking, preeclampsia, and gestational diabetes, might benefit the most from the combination method.

References:

Zaitsu, M.; Yoshihara, T.; Nakai, H.; Kubota, S. Optimal thermal control with sufficient nutrition may reduce the incidence of neonatal jaundice by preventing body-weight loss among NLBW infants not admitted to neonatal intensive care unit. Neonatology 2018, 114, 348-354.

Kubota, S.; Koyanagi. T.; Hori, E.; Hara, K.; Shimokawa, H.; Nakano, H. Homeothermal adjustment in the immediate postdelivered infant monitored by continuous and simultaneous measurement of core and peripheral body temperatures. Biol Neonate 1988, 54, 79-85.

Yoshimura, T.; Tsukimori, K.; Wake, N.; Nakano, H. The influence of thermal environment on pulmonary hemodynamic acclimation to extrauterine life in normal full-term neonates. J Perinat Med 2007, 35, 236-240.

Maisels, M.J.; Kring, E. Length of Stay, Jaundice, and Hospital Readmission. Pediatrics 1998, 101, 995-998.

Point 4: The time to regain birth weight is short, this points to the fact that most babies were probably predominantly formula-fed. Weight changes with exclusive breastfeeding have been described recently (in the Kaiser Permanente Healthcare Network;  Weight Change Nomograms for the First Month After Birth. Paul IM, et al  Pediatrics. 2016 Dec;138(6).

Response 4: We thank the reviewer for the comment. Paul et al. (2016) described neonatal weight changes with exclusive breastfeeding, suggesting that 50% of neonates may recover their weight around 10 days after birth [Paul 2016]. By contrast, in our study population, most of the neonates recovered their weight by 4 days after birth, suggesting that our nutrition regulation with supplemental formula milk may successfully meet the basal metabolism in our high-risk population. We added this discussion as follows:

Discussion, L276:

(Revised, revised parts are underlined): “Additionally, in a recent study with exclusively breastfed neonates, 50% of neonates recovered their weight around 10 days after birth [Paul 2016]. By contrast, most of the neonates recovered their weight by 4 days after birth in our study, suggesting that our nutrition regulation with supplemental formula milk may successfully meet the basal metabolism in our high-risk population.

References:

Paul, I.M.; Schaefer, E.W.; Miller, J.R.; Kuzniewicz, M.W.; Li, S.X.; Walsh, E.M.; Flaherman, V.J. Weight Change Nomograms for the First Month After Birth. Pediatrics 2016, 138, 10.1542/peds.2016-2625.

Point 5: As the authors point out it is not clear if this policy of early introduction of (partial?) formula feeding resulted in reduced numbers of breastfeeding at one month or later. This current study could be added (Flaherman V et al. J Pediatr. 2018 May;196:84-90.) for discussion, which also discusses readmission rates.

Response 5: We thank the reviewer for the comment. We added the discussion for early introduction of limited formula feeding, citing the article as follows:

Discussion, L324:

(Revised, revised parts are underlined): “In addition, a recent randomized clinical trial concluded that supplement breastfeeding with early limited formula milk (10 mL) did not interfere with breastfeeding [Flaherman 2018].

References:

Flaherman, V.J.; Narayan, N.R.; Hartigan-O'Connor, D.; Cabana, M.D.; McCulloch, C.E.; Paul, I.M. The Effect of Early Limited Formula on Breastfeeding, Readmission, and Intestinal Microbiota: A Randomized Clinical Trial. J. Pediatr. 2018, 196, 84-90.e1.

Point 6: The cost-effectiveness of this model of care should also be put into context. In some jurisdictions a community midwifery programme may supplement a model of early discharge (Flaherman V et al  Health Care Utilization in the First Month After Birth and Its Relationship to Newborn Weight Loss and Method of Feeding. Acad Pediatr. 2018 Aug;18(6):677-684).

Response 6: We thank the reviewer for the comment. In a previous cohort comprised with ~160,000 neonates at one of 14 Kaiser Permanente Northern California hospitals, the incidence of readmission was 0.5 times less prevalent in formula-fed neonates compared with breastfed neonates (2.1% vs. 4.3%), suggesting a potential savings of USD 7.8 million if the readmission rate of exclusively breastfed neonates approximated that of exclusively formula-fed neonates [Flaherman 2018]. In our cohort, the incidence of phototherapy was 0.3%, which approximates 0.07 times of that in exclusively breastfed neonates of the cohort of Kaiser Permanente Northern California hospital. Therefore, with our combination method, a potential savings would be USD 7.8*(0.5/0.07) = 55 million in that setting. In addition, improved education of clinicians/nurses and parents, as well as systemic institutional/community-wide approaches to manage neonatal jaundice, may also reduce unnecessary medical expenditure [Flaherman 2018; Bhutani 2006]. We added this discussion in the text, accordingly as follows:

Discussion, L336:

(Revised, revised parts are underlined): “Even in our high-risk study population, we observed a very low incidence of neonatal jaundice (0.4%), which corresponds to the incidence in Western settings [Maisels 1998; Eggert 2006]. Given that very low incidence of neonatal jaundice, a potential reduction of medical expenditures would be expected by the combination method, in addition to improved education (clinicians, nurses, and parents) and systemic institutional/community-wide approaches [Flaherman 2018; Bhutani 2006].

References:

Flaherman, V.; Schaefer, E.W.; Kuzniewicz, M.W.; Li, S.X.; Walsh, E.M.; Paul, I.M. Health Care Utilization in the First Month After Birth and its Relationship to Newborn Weight Loss and Method of Feeding. Acad. Pediatr. 2018, 18, 677-684.

Bhutani, V.K.; Johnson, L.H.; Schwoebel, A.; Gennaro, S. A Systems Approach for Neonatal Hyperbilirubinemia in Term and Near-Term Newborns. J. Obstet. Gynecol. Neonatal Nurs. 2006, 35, 444-455.

Eggert, L.D.; Wiedmeier, S.E.; Wilson, J.; Christensen, R.D. The Effect of Instituting a Prehospital-Discharge Newborn Bilirubin Screening Program in an 18-Hospital Health System. Pediatrics 2006, 117, e855-62.

Maisels, M.J.; Kring, E. Length of Stay, Jaundice, and Hospital Readmission. Pediatrics 1998, 101, 995-998.